# The aetiological relationship between depressive symptoms and health-related quality of life: A population-based twin study in Sri Lanka

**Panagiota Triantafyllou**[1]*, **Zeynep Nas**[1], **Helena M. S. Zavos**[2], **Athula Sumathipala**[3,4], **Kaushalya Jayaweera**[3], **Sisira H. Siribaddana**[5], **Matthew Hotopf**[6,7], **Stuart J. Ritchie**[1], **Frühling V. Rijsdijk**[1,8]

1 Social, Genetic & Developmental Psychiatry Centre, Institute of Psychiatry, Psychology and Neuroscience, King's College London, London, United Kingdom, 2 Department of Psychology, Institute of Psychiatry, Psychology and Neuroscience, King's College London, London, United Kingdom, 3 Institute for Research and Development, Colombo, Sri Lanka, 4 Research Institute for Primary Care and Health Sciences, Faculty of Health, Keele University, Keele, United Kingdom, 5 Faculty of Medicine & Allied Sciences, Rajarata University of Sri Lanka, Anuradhapura, Sri Lanka, 6 Psychological Medicine Department, Institute of Psychiatry, Psychology, and Neuroscience, King's College London, London, United Kingdom, 7 NIHR Biomedical Research Centre for Mental Health at the South London and Maudsley NHS Foundation Trust, King's College London, London, United Kingdom, 8 Psychology Department, Faculty of Social Sciences, Anton de Kom University, Paramaribo, Suriname

* panagiota.triantafyllou@kcl.ac.uk

**Data Availability Statement:** All relevant data are within the paper and its Supporting Information files.

## Abstract

### Objective

Depression often co-occurs with poor health-related quality of life (HRQL). Twin studies report genetic and individual-level environmental underpinnings in the aetiology of both depression and HRQL, but there is limited twin research exploring this association further. There is also little evidence on sex differences and non-Western populations are underrepresented. In this paper we explored the phenotypic and aetiological relationship between depressive symptoms and HRQL and possible sex differences in a low-middle-income Sri Lankan population.

### Method

Data for 3,948 participants came from the Colombo Twin and Singleton Follow-up Study (CoTaSS-2). Using self-report measures of depressive symptoms and HRQL, we conducted univariate and bivariate sex-limitation twin analyses.

### Results

Depressive symptoms showed moderate genetic (33%) and strong nonshared environmental influences (67%). Nonshared environment accounted for the majority of variance in all the subscales of HRQL (ranging from 68 to 93%), alongside small genetic influences (ranging from 0 to 23%) and shared environmental influences (ranging from 0 to 28%). Genetic

**Funding:** The Colombo Twin and Singleton study was funded in whole, or in part, by the Wellcome Trust [Grant number 093206/Z/10/Z], https://wellcome.org. For the purpose of open access, the author has applied a CC BY public copyright licence to any Author Accepted Manuscript version arising from this submission. FR, AS, SS and MH were involved in the second CoTaSS study as co-applicants of a Welcome trust funded grant. The funders had no role in study design, data collection and analysis, decision to publish, or preparation of the manuscript.

**Competing interests:** The authors have declared that no competing interests exist.

influences were significant for emotional wellbeing (23%). Shared environmental influences were significant for four out of the eight HRQL variables (ranging from 22–28%), and they were more prominent in females than males. Depressive symptoms were significantly associated with lower HRQL scores. These correlations were mostly explained by overlapping nonshared environmental effects. For traits related to emotional functioning, we also detected substantial overlapping genetic influences with depressive symptoms.

## Conclusions

Our study confirmed previous findings of a negative association between depressive symptoms and HRQL. However, some of the aetiological factors of HRQL differed from Western studies, particularly regarding the effects of shared environment. Our findings highlight the importance of cross-cultural research in understanding associations between psychological wellbeing and HRQL.

## Introduction

Health-related quality of life (HRQL) is a concept which describes the impact of perceived health status on daily social and physical functioning [1]. The inclusion of HRQL measures in healthcare provides a more person-centred evaluation of the impact of chronic physical conditions and effects of treatment [2]. Yet, even in the absence of physical problems, individuals may report somatic impairments, such as pain and fatigue, impairing their daily activities, which is the case with depression [3]. A similar pattern—between severity of broader depressive symptoms (i.e., low mood, loss of pleasure) and poor HRQL—has been observed in healthy, general population samples, suggesting that the absence of a diagnosis of a physical or emotional condition does not necessarily imply good health [3]. Studies using genetically sensitive designs have shown that the association between depressive symptoms and HRQL might in part be attributed to a shared genetic aetiology [4–7].

One method of exploring genetic aetiology is the twin design [8], which estimates the extent to which genetic and environmental factors explain the variance of a trait, or the covariance between multiple traits. Univariate twin studies in Western populations report that genetic influences explain 22–40% of the variance of depressive symptoms [9, 10]. The remaining variance is attributable to individual-level environmental influences. Traits encompassed in the broader concept of HRQL also show moderate heritability: life satisfaction (32%), emotional wellbeing (51%), self-reported health (40%), and bodily pain (30–68%) [11–14]. Unlike depressive symptoms, heritability for some HRQL traits increases in older age (i.e., self-reported health, pain) [13, 14], but this difference might reflect a vulnerability to age-related physical illness not yet presented in younger people. Unfortunately, there is limited twin research combining all the separate HRQL traits into a comprehensive HRQL measure. Still, heritability estimates from two studies in Denmark and in the USA using these measures are in line with studies exploring the variance of separate HRQL traits [15, 16]. Notably, the reported influence of shared environment in both depressive symptoms and HRQL is small and mostly non-significant, suggesting that the family environment in Western cultures has a negligible role in explaining the variance in mental and physical health.

In terms of association between depression and HRQL, a large Norwegian study reported a negative correlation between life satisfaction and major depression as assessed by the DSM-IV

($r$ = -.36). Genetic factors accounted for 73% of the phenotypic correlation [4]. Similarly, studies conducted in Europe, the UK, and the USA, exploring pain and fatigue, also report substantial genetic correlations with depression ($r_g$ = .36 to .71) [5, 6]. In the general, non-clinical population, half of the total phenotypic correlation between emotional wellbeing and depressive symptoms (r = -.55), has also been previously explained by genetic factors [7]. However, there is no research to our knowledge on the overlap of depressive symptoms with the different domains of HRQL.

Although sex differences are reported at mean level, that is, females consistently report more depressive symptoms and poorer HRQL status relative to males [2, 17], variance decomposition of depression [9, 10] and HRQL traits [11–16], as well as their covariance [4–7], does not show sex differences. The only sex differences in the aetiology of HRQL have been reported as differences between age groups [14, 15].

Research from non-Western populations, however, is still scarce. Non-Western populations show lower prevalence of depression and a more favourable state of HRQL than Western countries, although females remain disproportionally affected in terms of mental and physical health [17–20]. Regarding aetiology, twin findings from China and South Korea have found some heritability of depression in adolescent and young adult samples [21, 22], but there is limited research on representative samples in terms of age and socioeconomic status. A population-based study established in Sri Lanka, the Colombo Twin and Singleton study (*CoTaSS-1*) [23], has previously been used to explore the aetiology of various traits (i.e., emotional overeating, life events, pain, fatigue) and their overlap with depression and depressive symptoms [24–27]. One recent study using this South-Asian sample explored the aetiology of HRQL and its association with anxiety symptoms [28].

In the present study, we used a follow-up of the CoTaSS-1 study (*CoTaSS-2)* [29] to explore: i) the magnitude of genetic and environmental influences associated with depressive symptoms and HRQL; ii) the genetic and environmental overlap between depressive symptoms and HRQL; and iii) sex differences in these estimates. The importance of studying the common aetiology between depression and HRQL is to gain insight into the mechanisms of depression that may contribute to our understanding and aid development of therapeutic interventions and prevention strategies to reduce mental health as well as related quality of life.

## Methods and materials

### Sample

The CoTaSS-2 study [29] contains questionnaire data on aspects of physical health (i.e., chronic and metabolic disease), lifestyle (i.e., sleep, smoking), and psychopathology (i.e., depression, anxiety). Twin zygosity was identified via questionnaires on self-reported resemblance and on twin closeness (i.e., dressing alike, having common friends). All questionnaires have been translated and adapted to the Sri Lankan culture. Participants were mainly of Sinhalese ethnicity, but the sample was socioeconomically diverse (i.e., in terms of education, employment, marital status). Details of sociodemographic data can be found at Jayaweera et al. [23]. The sample consists of 2,921 twins (including incomplete twin pairs) and 1,027 singletons, aged from 19 to 91 years (mean age = 42.84, SD = 14.56). Sample composition is reported in **Table 1**. All participants provided written informed consent. Individuals that did not understand the consent process or the questionnaires due to language barriers were excluded. Participants that had successfully completed one or more study parts were offered 750 LKR (approximately £3.50 GBP) to compensate for their time. Ethical approval was obtained from the Faculty of Medical Sciences University of Sri Jayewardenepura Ethical Review Committee (USJP ERC) (reference number: 596/11) and from the Psychiatry, Nursing and Midwifery

**Table 1. Number of individuals per sex and zygosity.**

| Zygosity group | Males | Females | Total |
|---|---|---|---|
| **Monozygotic twins** | | | 1,263 |
| Paired twins* | 478 | 668 | |
| Single twins | 55 | 62 | |
| **Dizygotic twins** | | | 850 |
| Paired twins | 302 | 410 | |
| Single twins | 63 | 75 | |
| **Opposite sex twins** | | | 808 |
| Paired twins | 343 | 343 | |
| Single twins | 37 | 75 | |
| **Singletons** | 388 | 639 | 1,027 |
| **Total** | 1,676 | 2,272 | **3,948** |

*Paired twins: data for twins coming from a complete pair.

Research Ethics Subcommittee, King's College London, UK (reference number: PNM/10/11-124).

## Measures

**Depressive symptoms.** Depressive symptoms were assessed by the revised edition of the Beck Depression Inventory (*BDI-II*), which consists of 21 items each corresponding to a depressive symptom (i.e., pessimism, loss of pleasure, feelings of punishment), and assesses severity during the past two weeks [30]. The total score indicates depression severity: no or very low depression (0–13); mild depression (14–19); moderate depression (20–28); and severe depression (29–63). The Sinhalese version of the BDI-II has excellent internal consistency ($\alpha$ = 0.93) for screening for depressive symptoms in the Sri Lankan population [31].

**Health-related quality of life (HRQL).** HRQL was assessed by the Short Form Health Survey (*SF-36*) [32]. The 36 items are broken down into eight domains as follows: general health (6 items; i.e. perceptions of their overall state of health); emotional wellbeing (5 items; i.e. feeling emotionally well or anxious/depressed); social functioning (2 items; i.e. interference with social activities due to emotional and/or mental health problems); physical functioning (10 items; i.e. participation in physical activities); energy and fatigue (4 items; i.e. feeling energetic or tired); pain (2 items; i.e. limitations in activities due to bodily pain); role-physical (4 items; i.e. limitations in activities of their daily role due to physical problems); and role-emotional (3 items; i.e. limitations in activities of their daily role due to emotional problems). Scores range from 0 (poor health state) to 100 (most favourable health state). The SF-36 has been validated in the UK and Sri Lanka [17, 33] and has also high predictive validity for depression as demonstrated on patients with chronic physical illness and in the general population (specificity of 82–92%) [34, 35].

## Statistical analysis

Statistical tests were performed in R (*version 3.6.3*) [36] and genetic model-fitting analyses were conducted using the *OpenMx* package within R (*version 2.17.4*) [37].

We first conducted *t*-tests to obtain the mean differences between males and females in their scores for the BDI-II and for each of the eight SF-36 scales. As part of data processing in twin model fitting, sex and age were regressed out from all variables to prevent from increasing the correlations between twin pairs and potentially inflating the shared environment [38].

When necessary, variables were also log-transformed to reduce skew. The social functioning scale was excluded from this process because it was analysed as a categorical variable. Instead, the main effects of age and sex were incorporated as covariates on the thresholds within a liability model. The sample was split by sex and zygosity to test for sex differences and singletons were also included in all the analyses to help estimate the phenotypic correlations and variances.

## Genetic model fitting

Twin models decompose the variance of a trait or the covariance between multiple traits in terms of latent genetic and environmental factors. The logic of the twin model [8] is based on various assumptions, including: i) Identical (MZ) twins share 100% of segregating genes whereas for fraternal (DZ) twins this is only 50% on average (this means that the genetic factor *A* is correlated 1 and .5 respectively); ii) twins reared together share their environment to the same extent regardless of their zygosity (this means that the factor *C* is correlated 1 in both the MZ and DZ model); iii) the unique or individual-specific environment *E* explains the difference between twin siblings, and is uncorrelated across family members [39, 40].

Using this information in a structural equation modelling (SEM) framework, the univariate ACE twin model expresses the correlation of MZ twins as *A+C*, and that of DZ twins as *.5A +C*. The model is fitted to the observed correlational structure of the data to decompose the variance of a single trait into genetic influences *(A)*, shared environment *(C)*, and nonshared environment *(E)* (which also includes measurement error).

This can be extended to bivariate ACE analyses, estimating the ACE parameters for two traits separately, as well as the *A*, *C* and *E* correlation between the traits [41]. This means that we can partition the total implied phenotypic correlation (*Rph*) into *Rph-A*, *Rph-C*, and *Rph-E*, which represent the extent to which factors *A*, *C*, and *E* respectively contribute to the total overlap between the traits. The magnitude of the parameters is indicated by the correlation (r) ratio between MZ and DZ pairs of twins (*r*MZ:*r*DZ). Within-trait cross-twin *r*MZ:*r*DZ provides evidence for the most likely variance components of a trait (e.g. a 2:1 ratio indicating the effects of *A*; a 1:1 ratio indicating *C*). Similarly, cross-twin cross-trait *r*MZ:*r*DZ provide information about the nature of the covariance of two traits (e.g. a 2:1 and 1:1 ratio indicating *A* and *C* to be the main cause of overlap, respectively). When cross-twin cross-trait correlations are nonsignificant, then *E* is the most likely cause of the observed phenotypic correlation between two traits [8, 41].

Additionally, twin models allow testing for sex differences in the aetiology of a variable or on the relationship between multiple variables. These models can test for: quantitative sex differences, where the magnitude of the same genetic and environmental factors associated with the variances of the traits or their covariance differs across sexes; and qualitative sex differences, where the variance and covariance of the traits are associated with different genetic and environmental factors (indicated if the correlation in opposite-sex pairs is significantly lower than that of DZ same-sex pairs).

In addition to the models above, we have to rule out that sex differences show up due to absolute variance differences rather than aetiology. For example: A = 60, C = 20 and E = 20 in females and A = 6, C = 2 and E = 2 in males in both cases give the same standardized results. This is modelled by estimating the same A, C and E for males and females but allowing the variance for females to be multiplied by a constant to scale up the variance. Thus, prior to bivariate genetic modelling, for each variable we first examined sex differences in variance in a phenotypic correlation model, followed by three univariate ACE models: a quantitative heterogeneity ACE model; a scalar ACE model, where parameters are equated across sexes, but the

female variance is multiplied by a constant (scalar); and a homogeneity ACE model, where the parameters are equated and not allowed to vary. The bivariate ACE models were then fitted based on the outcome of the univariate ACE results: for example, if for one variable a scalar model was the best fit, and for the second one a quantitative heterogeneity ACE (or homogeneity ACE) was the best fit, we would run a bivariate 'hybrid' scalar-heterogeneity or 'hybrid' scalar-homogeneity ACE model. In addition, in the hybrid scalar-heterogeneity bivariate models we tested for qualitative sex differences in *A* and *C*. To determine best fit of nested models, the criterion of likelihood ratio chi-square ($\chi^2$) was used, a statistic calculated as the difference between minus two times the log of the likelihoods of the models (-2*$LL$). A significant *p*-value for the nested sub model suggests that the full model is a better fit to the data.

## Results

### Descriptive statistics

Overall mean BDI-II scores were low (4.85, $SD$ = 6.16); a minority of participants met the criteria for mild (5%), moderate (3%), and severe depression (1%). A Welch's *t*-test revealed that females reported significantly higher depressive symptoms (5.50, $SD$ = 6.59) than males (3.99, $SD$ = 5.46), (t (3790.4) = -7.73, $p$ < .001, $d$ = 0.25). SF-36 scores were high, indicating favourable HRQL overall (**Table 2**). Males scored significantly higher than females in the SF-36. **S1 Table** details individual t-tests for SF-36 variables.

### Constrained correlation model estimates

The MZ and DZ ratio (*r*MZ: *r*DZ) of within-trait cross-twin correlations (**Table 3**) provides information on the sources of variance in the traits. For depressive symptoms, *r*MZ were greater than *r*DZ suggesting genetic effects (*A*). For the majority of the HRQL variables, the *r*MZ; *r*DZ was approximately 1:1 for both sexes, suggesting greater contribution of shared environment (C). For males, the *r*DZ for some variables were not significant, and the *r*MZ was

**Table 2. Mean (SD) scores of SF-36 scales overall and split by sex.**

| Variable | Overall number of individuals | Overall mean (SD) | Number of Individuals by sex | | Mean (SD) by sex |
|---|---|---|---|---|---|
| **General Health** | 3856 | 60.84 (16.17) | M | 1621 | 62.42 (15.02) |
| | | | F | 2235 | 59.57 (16.87) |
| **Emotional Wellbeing** | 3856 | 78.15 (15.93) | M | 1622 | 79.96 (14.60) |
| | | | F | 2234 | 76.87 (16.70) |
| **Energy/Fatigue** | 3856 | 73.28 (16.93) | M | 1622 | 73.96 (16.40) |
| | | | F | 2234 | 72.80 (17.27) |
| **Pain** | 3857 | 86.67 (20.20) | M | 1621 | 88.70 (18.93) |
| | | | F | 2236 | 85.11 (21.05) |
| **Physical Functioning** | 3856 | 89.22(18.91) | M | 1620 | 92.33 (16.68) |
| | | | F | 2236 | 86.89 (20.08) |
| **Social Functioning** | 3840 | 89.32 (19.41) | M | 1615 | 90.27 (18.62) |
| | | | F | 2225 | 88.70 (19.89) |
| **Role Physical** | 3856 | 82.02 (35.67) | M | 1620 | 84.95 (33.31) |
| | | | F | 2236 | 79.64 (37.37) |
| **Role Emotional** | 3856 | 86.98 (31.01) | M | 1620 | 89.36 (28.32) |
| | | | F | 2236 | 85.01 (32.90) |

*M* = males, *F* = females

**Table 3. Cross-twin within-trait correlations for depressive symptoms and SF-36 scales in same and opposite sex twin pairs (with 95% CIs).**

| Variable | MZM | DZM | MZF | DZF | DZOS |
|---|---|---|---|---|---|
| **Depressive Symptoms** | **.31** | **.25** | **.35** | **.21** | .10 |
| | **(.15/.44)** | **(.08/.40)** | **(.25/.44)** | **(.07/.33)** | (-.01/.20) |
| **General Health** | **.24** | **.27** | **.30** | **.34** | .11 |
| | **(.11/.36)** | **(.10/.41)** | **(.20/.40)** | **(.22/.45)** | (.00/.22) |
| **Emotional Wellbeing** | **.24** | .17 | **.30** | **.22** | .11 |
| | **(.13/.35)** | (-.02/.34) | **(.19/.40)** | **(.10/.34)** | (.00/.21) |
| **Energy/Fatigue** | **.18** | **.21** | **.26** | **.27** | **.19** |
| | **(.05/.30)** | **(.02/.37)** | **(.15/.36)** | **(.14/.39)** | **(.08/.29)** |
| **Pain** | .11 | **.36** | **.17** | **.22** | .05 |
| | (-.02/.24) | **(.18/.50)** | **(.06/.27)** | **(.10/.33)** | (-.07/.16) |
| **Physical Functioning** | .12 | **.47** | **.38** | **.23** | **.22** |
| | (-.13/.32) | **(.23/.61)** | **(.28/.47)** | **(.09/.36)** | **(.09/.33)** |
| **Social Functioning** | **.42** | **.43** | **.43** | **.49** | **.32** |
| | **(.22/.60)** | **(.16/.65)** | **(.27/.57)** | **(.28/.66)** | **(.15/.47)** |
| **Role Physical** | .06 | .19 | **.26** | **.28** | .02 |
| | (-.10/.20) | (-.02/.37) | **(.14/.36)** | **(.16/.40)** | (-.08/.12) |
| **Role Emotional** | **.19** | .09 | **.18** | **.32** | .06 |
| | **(.05/.31)** | (-.13/.30) | **(.07/.29)** | **(.19/.44)** | (-.04/.17) |

Note: *MZM*: monozygotic males; *MZF*: monozygotic females; *DZM*: dizygotic males; *DZF*: dizygotic females; *DZOS*: dizygotic opposite sex twins. Significant correlations are shown in bold.

generally larger than the $r$DZ, suggesting greater contribution of genetics. In females, all within-trait cross-twin correlations were significant. For some variables, the $r$DZ was non-significantly smaller than the $r$MZ across sex, suggesting negligible effects of familial factors (A and C). All within-twin cross-trait phenotypic correlations were negative and significant ($r$ = -.29 to -.57) (**S2 Table**). The nature of this overlap can be inferred by the cross-trait cross-twin ratio $r$MZ:$r$DZ (**S3 Table**). The majority of the cross-trait cross-twin $r$DZ were non-significant for males. For most variables, the $r$MZ:$r$DZ ratio approached 1:1 across sex, suggesting shared environmental effects in their covariance with depressive symptoms.

## Univariate variance decomposition

For depressive symptoms, emotional wellbeing, pain, and the energy/fatigue scale there were no sex differences in aetiology. For general health, role limitations due to physical problems, role limitations due to emotional problems, and social functioning sex differences in aetiology were found. **S4 Table** details the univariate model fit statistics, selection and justification of the best-fitting model. In the bivariate model, when depressive symptoms is analysed with the HRQL scale variables, the knowledge we have from the univariate models (in terms of best-fit) is used to model the combined data.

Standardized variance components from the univariate analyses are presented in **Table 4**. Depressive symptoms showed moderate but significant genetic influences (33%, [95% CI = 13–40%]) and strong nonshared environmental influences (67%, [95% CI = 60–75%]). For the eight SF-36 scales, the majority of variance was attributed to nonshared environment (68–93%). Genetic influences were moderate and significant for emotional wellbeing only (23%, [95% CI = 2–34%]), for males and females. For the energy/fatigue scale we found moderate and significant influences from shared environment (21%, [95% CI = 5–28%]). For

**Table 4. Variance decomposition (univariate ACE estimates) for depressive symptoms and SF-36 scales.**

| Variable | Sex | A (95% CI) | C (95% CI) | E (95% CI) |
|---|---|---|---|---|
| *Depressive Symptoms | M\|F | **.33 (.13/.40)** | .00(.00/.15) | **.67(.60/.75)** |
| General Health | M | .17 (.00/.36) | .10(.00/.29) | **.73 (.62/.85)** |
| | F | .04 (.00/.30) | **.28 (.05/.39)** | **.68 (.59/.76)** |
| Social Functioning | M | .07 (.00/.23) | .00 (.00/.00) | **.93 (.76/.99)** |
| | F | .02 (.00/.18) | **.23 (.09/.33)** | **.75 (.67/.84)** |
| Role limitations Physical problems | M | .09 (.00/.23) | .00 (.00/.15) | **.91 (.77/1)** |
| | F | .00 (.00/.31) | .27 (.00/.35) | **.73 (.64/.82)** |
| Role limitations Emotional problems | M | .17 (.00/.30) | .00 (.00/.16) | **.82 (.69/.95)** |
| | F | .00(.00/.16) | **.23(.08/.31)** | **.77 (.69/.85)** |
| *Emotional Wellbeing | M\|F | **.23 (.02/.34)** | .04 (.00/.21) | **.73 (.66/.81)** |
| * Pain | M\|F | .00 (.00/.00) | .15 (.00/.21) | **.85 (.79/.90)** |
| *Physical Functioning | M\|F | .16 (.00/.39) | .16 (.00/.33) | **.68 (.60/.77)** |
| **Energy/Fatigue | M\|F | .01 (.00/.23) | **.22(.05/.28)** | **.77 (.70/.83)** |

Note: Significant parameters are indicated in bold (95%CI not including zero).

general health, limitations due to physical problems, limitations due to emotional problems, and social functioning, shared environment was moderate and significant in females only. No significant familial influences (A or C) were detected in males.

Variables with (*) and (**) have no male/female breakdown, because they were fit to a model where the standardised ACE parameters are equated across sexes, and either allow for variances to differ by a constant multiplier (*scalar), or not (**homogeneity model). $M$ = males, $F$ = females, $A$ = additive genetic influences, $C$ = shared environmental influences, $E$ = nonshared environmental influences.

## Covariance decomposition between depressive symptoms and HRQL

The bivariate ACE models were specified based on the best-fitting univariate models, some-times resulting in fitting different types of models for depressive symptoms and the scale variables (hybrid models) which are detailed in S5 Table. Covariance components from the bivariate analyses are presented in **Table 5**.

All phenotypic correlations between depressive symptoms and HRQL were negative and significant ($Rph$ = -.28 to -.56) (**S3 Table**), suggesting that depressive symptoms are associated with lower HRQL. The majority of this phenotypic relationship was explained by correlated nonshared environmental factors ($Rph$-$E$ = -.12 to -.36).

Aetiologically, we found a significant negative genetic correlation between depressive symptoms and emotional wellbeing ($Ra$ = -.90, 95% CI = -1.00/-.52), suggesting that 41% of the phenotypic correlation ($Rph$ = -.56) was due to overlapping genetic influences ($Rph$-$A$ = -.23). This was also the case for depressive symptoms and the energy/fatigue scale ($Rph$-$A$ = -.20), although the model did not converge to estimate the 95% CI accurately for the corresponding aetiological correlation ($Ra$). No significant influences from familial factors were detected for the remaining variables (pain, physical functioning). **S6 Table** details the genetic ($Ra$), shared environmental ($Rc$), and non-shared environmental ($Re$) correlations between depressive symptoms and the HRQL traits.

For the analysis of depressive symptoms with scale variables that show sex differences in aetiology model, we explored sex differences in overlap. Model comparisons are detailed in **S5 Table**. For the correlated nonshared environment ($Re$), point estimates were lower for females

**Table 5. ACE decomposition of the phenotypic correlations between SF-36 scales and depressive symptoms.**

| Variable | Sex | Total Rph (95% CI) | Rph-A (95% CI) | Rph-C (95% CI) | Rph-E (95% CI) |
|---|---|---|---|---|---|
| **General Health** | M | **-.30** | -.10 | **.06** | -.27 |
| | | **(-.34/-.25)** | (-.20/.00) | **(.02/.12)** | **(-.36/-.18)** |
| | F | **-.29** | **-.16** | .02 | **-.14** |
| | | **(-.32/-.25)** | **(-.27/-.02)** | (-.11/.11) | **(-.21/-.08)** |
| **Social Functioning** | M | **-.45** | -.20 | -02 | -.23 |
| | | **(-.49/-.40)** | (-.34/.01) | (-.23/.06) | **(-.33/-.13)** |
| | F | **-.46** | -.16 | -.04 | **-.26** |
| | | **(-.50/-.41)** | (-.32/.04) | (-.19/.08) | **(-.35/-.18)** |
| **Role Physical** | M | **-.31** | -.12 | .05 | -.24 |
| | | **(-.35/-.26)** | (-.20/.00) | (-.01/.10) | **(-.33/-.15)** |
| | F | **-.31** | -.07 | **-.12** | -.12 |
| | | **(-.35/-.27)** | (-.16/.02) | **(-.18/-.05)** | **(-.19/-.05)** |
| **Role Emotional** | M | **-.45** | **-.11** | .00 | -.34 |
| | | **(-.49/-.41)** | **(-.21/-.02)** | (-.02/.05) | **(-.43/-.26)** |
| | F | **-.44** | **-.18** | -.04 | -.23 |
| | | **(-.48/-.41)** | **(-.26/-.10)** | **(-.09/-.02)** | **(-.30/-.16)** |
| *Emotional Wellbeing | M\|F | **-.56** | **-.23** | .03 | -.36 |
| | | **(-.59/-.54)** | **(-.30/-.08)** | (-.08/.06) | **(-.43/-.31)** |
| **Energy/Fatigue | M\|F | **-.44** | **-.20** | .02 | **-.26** |
| | | **(-.46/-.41)** | **(-.27/-.08)** | (-.08/.07) | **(-.32/-.21)** |
| *Pain | M\|F | **-.35** | -.14 | -.01 | -.20 |
| | | **(-.38/-.32)** | (-.23/.02) | (-.12/.05) | **(-.26/-.15)** |
| *Physical Functioning | M\|F | **-.28** | -.11 | -.05 | **-.12** |
| | | **(-.31/-.25)** | (-.23/.05) | (-.16/.05) | **(-.18/-.06)** |

Note: Significant parameters are indicated in bold (95%CI not including zero).

Variables with (*) and (**) have no male/female breakdown, because they were fit to a model where the standardised ACE parameters are equated across sexes, and either allow for variances to differ by a constant multiplier (*scalar model), or not (**homogeneity model). M = males, F = females, A = additive genetic influences, C = shared environmental influences, E = nonshared environmental influences. Rph = total phenotypic correlation between SF-36 and depressive symptoms; *Rph-A*, *Rph-C*, *Rph-E* = phenotypic correlation due to correlated A, C, and E respectively.

than males in some variables (general health and limitations due to emotional problems), but the 95% CI were overlapping. There was evidence of familial effects on the phenotypic correlation (*Rph-A*, *Rph-C*) in all variables except social functioning, but the model did not converge to estimate the 95% CI accurately for the corresponding aetiological correlations (*Ra*, *Rc*). We did however find a significant genetic correlation between depressive symptoms and limitations due to emotional problems for males only (*Ra* = -.47, 95%CI = -.84/ -.13), suggesting that 24% of the total phenotypic correlation (*Rph* = -.45) was due to genetic influences (*Rph-A* = -.11).

## Discussion

This is the first study, to our knowledge, to explore both the aetiology and covariance between depressive symptoms and HRQL in a low-to-middle income non-Western population. We used a comprehensive, validated measure of HRQL, the SF-36 (tapping into eight domains), and a continuous measure of depressive symptoms, both translated and adapted in accordance with the Sri Lankan languages and culture.

Overall levels of depressive symptoms were lower than estimates from Western populations and females were disproportionally affected, consistent with cross-cultural studies [19, 20] and

with previous studies using the CoTaSS sample [23, 27]. HRQL ratings were generally higher in our sample than in the UK [17]. Consistent with Western populations [17, 19], females rated their mental and physical health less favourably than males.

## Aetiology

Univariate model-fitting results revealed significant and substantial heritability for depressive symptoms in both sexes (33%) and the remaining variance was explained by the nonshared environment, which is consistent with Western studies [9, 10]. This is in contrast to previous reports using the CoTaSS dataset [25, 26], where reported sex differences were due to unaccounted unequal variances across sex, rather than aetiology. Fitting a model with the same A, C and E parameters and allowing the female variance to be a scalar bigger resulted in a better fitting (more parsimonious) model.

For HRQL, individual differences were largely explained by nonshared environmental influences (68–93%), alongside small influences from familial factors (22–28%). In both sexes, we found significant genetic influences in the variance of emotional wellbeing (23%), and significant influences from the shared environment in the variance of energy/fatigue (22%). In females only, shared environment was significant for some traits (23–28%).

A univariate twin study from the US that used the SF-36 reported similar heritability for the emotional wellbeing scale [15], but the sample consisted exclusively of middle-aged males. Studies including both sexes of different age groups report higher heritability for traits related to emotional wellbeing (32–51%), but the explored phenotypes were broader (i.e., life satisfaction, subjective wellbeing) [12–14]. Nevertheless, our findings are comparable and indicate that, similar to the West, genetic factors might increase the risk for poor emotional wellbeing in both males and females.

A striking difference with Western studies was the substantial shared environmental component, which was especially pronounced in females. Cultural effects might explain why influences from shared environment have not been observed in the West. Sri Lanka is classed as a collectivist society where tradition, extended family members, and the elders are particularly valued [42, 43]. Therefore, influences that are associated with the environment shared between the twins (i.e., parental beliefs) might be increased in our sample comparing to Western societies which are more individualistic. Sri Lanka is also within the lowest ranking positions worldwide in terms of female economic participation and opportunity [43], and it has been previously reported that employment rates are more than double in males than in females in our sample [24]. The increased focus in the home (i.e., domestic work, providing support to the family and the parents) and the lack of economic independence in females could increase the role of the shared environment in comparison to males [25–27]. Therefore, for females in Sri Lanka in particular, the family environment might play a more important role in their mental and physical health than in other countries.

## Phenotypic and aetiological correlations

Bivariate model fitting results showed significant negative correlations between all dimensions of HRQL and depressive symptoms ($Rph$ = -.28 to -.56), suggesting that factors that increase vulnerability to depressive symptoms also exert negative effects on self-reported HRQL. The phenotypic correlations were mostly explained by nonshared environment ($Rph$-$E$ = -.12 to -.36), alongside evidence of small overlapping genetic effects ($Rph$-$A$ = -.07 to -.23). The correlated nonshared environmental influences were negative and large, suggesting a major role of individual factors that increase depressive symptoms and affect HRQL negatively. Those influences were greater in males than in females. Apart from physical and emotional functioning in

females, shared environmental correlations with depressive symptoms were largely non-significant, suggesting negligible overlapping influences of the social or family environment. We found significant negative genetic correlations between depressive symptoms with role of emotional functioning and emotional wellbeing variables, indicating that there is a likelihood of a partially shared genetic liability with these traits.

Comparable research in the West is limited, but two bivariate twin studies also report negative phenotypic correlations between depressive symptoms and emotional wellbeing (-.31 to -.55), most of which were accounted for by significant genetic factors [4, 7]. Our study therefore indicates comparable aetiology with the West. Together with the univariate estimates, this suggests that some heritable factors that increase the risk for depression also increase the risk for poor emotional functioning, yet both traits are also affected by heritable factors that do not overlap.

The substantial contribution of nonshared environment in psychological and physical health is in line with Western studies, and with previous studies using the CoTaSS sample [24–27]. Nonshared environmental factors typically refer to individual-level factors that are unique to each twin, such as lifestyle (i.e., smoking, diet), life events (i.e., accidents, divorce), or chronic illness. However, this definition mostly concerns developed countries. It is possible that the nature of nonshared environmental influences might be different in this population compared to the West. Sri Lanka has been impacted by a long-running civil war (1983–2009) and a Tsunami (2004), with thousands of deaths and devastating effects in the economy [23, 42]. Although the Colombo district was not directly impacted by these events, parts of the population might have been disproportionally affected, either socioeconomically or emotionally (i.e., bereavement). Such pervasive experiences might lead to more environmental variability within a population, and consequently be captured by the nonshared environment [25]. This environmental variability might also explain why the heritability of most HRQL traits was non-significant. It must be noted though that the large contribution of the nonshared environment also includes measurement error.

## Strengths and limitations

First, we cannot exclude the possibility of response bias in the questionnaires, considering that mental health problems still carry social stigma in Sri Lanka [44]. Second, although the SF-36 has good discriminant validity from depression [34], we cannot exclude some overlap between the items of the BDI-II and the emotional scales of the SF-36. Third, the genetic models did not converge in finding 95% CI for some aetiological correlations. This lack of precision is associated with low statistical power and has been reported in previous analyses of this sample when testing for sex differences [25, 26]. This may mean that quantitative and qualitative sex differences could have been missed. Power constraints would also not make it possible to test for age effects by splitting the sample by age group, in combination with sex and zygosity. Lastly, our results provide evidence from a South Asian population, but replication using large samples from other low-to-middle income populations is needed for the findings to be generalised.

Considering the limited twin research in low-to-middle populations however, our study extends the literature in mental and physical health. Our sample was diverse in terms of age and socioeconomic factors, and we included singletons in our analyses, increasing generalisability of our results to the wider population [8]. Also, we used a continuous measure of depressive symptoms which was more appropriate to capture the broader spectrum in the general population than categorical measures.

### Future directions

Our findings indicate modest associations between severity of depressive symptoms and lower psychological and physical functioning, largely due to overlapping genetic and environmental influences. Previous studies, however, propose that these influences might be time-dependent, particularly for traits associated with health-related status [12, 15]. Further studies adopting a longitudinal design could explore these effects. Moreover, the association between depressive symptoms and HRQL provides no evidence for the direction of causality. It would be worthwhile exploring whether possible causal effects from depressive symptoms to poor HRQL and vice versa are plausible. Still, temperamental characteristics, that are strongly influenced by genetics, might exert effects on both physical and emotional wellbeing, as well as on the non-shared environment [45]. For example, higher scores in pessimism could increase the impact of poverty on an individual. Such additional factors should be controlled when trying to infer directionality.

Our sample was made up of a healthy South-Asian population, indicating the role of risk factors that can increase the vulnerability to both psychological and physical illness. For depressive symptoms and emotional functioning, the substantial heritability that we observed suggests that therapeutic interventions might be more effective when they are combined with greater awareness surrounding heritability of mental health. Raising awareness might also reduce the stigma associated with mental illness in Sri Lanka, improving mental health services and treatment seeking [44]. Given the substantial influence of the shared environment, preventative interventions might be especially effective for females if incorporated appropriately. Identifying the nature of individual-level environmental exposures could also improve person-centred therapeutic interventions to reduce mental and physical health decline.

## Conclusion

As in the West, individual differences in HRQL in Sri Lanka result from a combination of genetic and environmental factors that are shared with depressive symptoms, as well as factors that are specific in the aetiology of HRQL. However, we found substantial influences of shared environment particularly in females which were not observed in the West, highlighting the importance of cross-cultural approaches to health. Our findings extend the literature on mental and physical illness in non-Western populations which has not been extensively explored and form the basis for future investigation of two questions: the direction of causation between depressive symptoms and poor HRQL, and the nature of environmental influences on their aetiology and covariance.

## Supporting information

**S1 Table. Sex differences in means for SF-36.**
(DOCX)

**S2 Table. Within-twin cross-trait correlations.**
(DOCX)

**S3 Table. Cross-twin cross-trait correlations.** *MZM*: monozygotic males; *MZF*: monozygotic females; *DZM*: dizygotic males; *DZF*: dizygotic females; *DZOS*: dizygotic opposite sex twins. Significant correlations are shown in bold.
(DOCX)

**S4 Table. Univariate ACE model-fit statistics.** The best fitting models are indicated in bold. *Sat*: saturated phenotypic correlation model; *Sub1*: constrained correlation model; *HetACE*:

quantitative heterogeneity model testing for quantitative sex differences in additive genetic variance *(A)*, shared environment variance *(C)*, and non-shared environment variance (E); *ScACE*: scalar model, model where variances are allowed to differ across sexes by a constant multiplier; *HomACE*: homogeneity model, where the standardised ACE parameters are equated across sex; *-2LL*: minus twice the log of the likelihood of the data; *df*: degrees of freedom; *ΔLL(Δdf)*: the difference in -2LL and df of two models which is $\chi^2$ distributed. *AIC*: Akaike's Information Criterion.
(DOCX)

**S5 Table. Bivariate ACE model-fit statistics.** The best fitting models are indicated in bold. *Sat* = saturated phenotypic model; *Scalar* = model where variances are allowed to differ across sexes by a constant multiplier; *Scalar HetACE* = hybrid scalar-heterogeneity ACE model testing for quantitaive sex differences only by accounting for sex variance differences in depressive symptoms (scalar variable); *Scalar QualA*, *Scalar QualC* = hybrid scalar-heterogeneity ACE models testing for quallitative sex differences in additive genetic variance *(A)* and shared environment variance *(C)* respectively, by accounting for sex variance differences in depressive symptoms; *HomACE* = homogeneity model where the standardised ACE parameters are equated across sex; *Scalar HomACE* = hybrid scalar-homogeneity ACE model where the standardised ACE parameters are equated across sex, allowing for sex variance differences in depressive symptoms. The Scalar QualA, Scalar QualC, and HomACE models were compared to the Scalar HetACE model. For variables with a Scalar HomACE or a Scalar model only, we did not fit heterogeneity models because there were no aetiological differences across sexes identified in the univariate analyses. *-2LL*: minus twice the log of the likelihood of the data; *df*: degrees of freedom; *ΔLL(Δdf)*: the difference in -2LL and df of two models which is $\chi^2$ distributed. *AIC*: Akaike's Information Criterion.
(DOCX)

**S6 Table. Genetic (*Ra*) and environmental (*Rc, Re*) correlations.** Note: (1/1) and (-1/-1): model did not converge to estimate the 95%CI; Variables fit to a scalar-scalar model (*) or a scalar-homogeneity model (**) do not have a male/female breakdown because the (standardized) ACE parameters are equated across sexes, and either the variances are allowed to differ by a constant multiplier (scalar), or not (homogeneity model); M = Male, F = Female.
(DOCX)

**S7 Table. Data dictionary CoTaSS.**
(DOCX)

**S1 File. CoTaSS depression-SF36 Data.**
(ZIP)

## Acknowledgments

We are grateful to the participants of the Colombo Twin and Singleton Study for their participation in this research.

## Author Contributions

**Conceptualization:** Panagiota Triantafyllou, Zeynep Nas, Stuart J. Ritchie, Frühling V. Rijsdijk.

**Formal analysis:** Panagiota Triantafyllou, Zeynep Nas.

**Funding acquisition:** Helena M. S. Zavos, Athula Sumathipala, Kaushalya Jayaweera, Sisira H. Siribaddana, Matthew Hotopf, Frühling V. Rijsdijk.

**Investigation:** Helena M. S. Zavos, Athula Sumathipala, Kaushalya Jayaweera, Sisira H. Siribaddana, Matthew Hotopf, Frühling V. Rijsdijk.

**Methodology:** Zeynep Nas, Helena M. S. Zavos, Frühling V. Rijsdijk.

**Resources:** Zeynep Nas, Helena M. S. Zavos, Athula Sumathipala, Kaushalya Jayaweera, Sisira H. Siribaddana, Matthew Hotopf, Frühling V. Rijsdijk.

**Supervision:** Zeynep Nas, Stuart J. Ritchie, Frühling V. Rijsdijk.

**Writing – original draft:** Panagiota Triantafyllou, Zeynep Nas, Frühling V. Rijsdijk.

**Writing – review & editing:** Zeynep Nas, Helena M. S. Zavos, Athula Sumathipala, Kaushalya Jayaweera, Sisira H. Siribaddana, Matthew Hotopf, Stuart J. Ritchie, Frühling V. Rijsdijk.

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
