## [Decision Letter · Decision Letter 0]

23 Dec 2021

PONE-D-21-28109The aetiological relationship between depressive symptoms and health-related quality of life: A population-based twin study in Sri LankaPLOS ONE

Dear Dr. Triantafyllou,

Thank you for submitting your manuscript to PLOS ONE. After careful consideration, we feel that it has merit but does not fully meet PLOS ONE’s publication criteria as it currently stands. Therefore, we invite you to submit a revised version of the manuscript that addresses the points raised during the review process.

I was fortunate to receive 2 excellent reviews of your manuscript. Please address all the comments raised when revising your paper. In addition, I would be grateful if you could address the following points from my own reading: 

Abstract Results: The quoted figures for depression add up to 100%, for HRQL, a range is given for E and a combined range for A and C. It is not clear in abstract what the ranges are: confidence intervals? Sub-scales? I recommend separating out the A and C estimates. I am not sure how they can add up to 100% with the ranges given either. For example if the E estimate is at the upper range, should there be a lower range value for A and C to keep the total to 100%?

Rph-A etc abbreviations needs explanation for this journal as you give in the method. That won’t help readers at the abstract stage, however. I recommend leaving these figures out of the abstract and just explaining conceptually what these results mean here.

Introduction

Final paragraph. Check numbering.

Method

Genetic Model fitting: Please add a reference supporting the validity of the twin modelling  assumptions mentioned.

Discussion

Para 1 Please comment on why the results of this study may differ from other analyses of the same dataset.

 Please submit your revised manuscript by Feb 06 2022 11:59PM. If you will need more time than this to complete your revisions, please reply to this message or contact the journal office at plosone@plos.org. Please include the following items when submitting your revised manuscript:A rebuttal letter that responds to each point raised by the academic editor and reviewer(s). You should upload this letter as a separate file labeled 'Response to Reviewers'.A marked-up copy of your manuscript that highlights changes made to the original version. You should upload this as a separate file labeled 'Revised Manuscript with Track Changes'.An unmarked version of your revised paper without tracked changes. You should upload this as a separate file labeled 'Manuscript'.If applicable, we recommend that you deposit your laboratory protocols in protocols.io to enhance the reproducibility of your results. Protocols.io assigns your protocol its own identifier (DOI) so that it can be cited independently in the future. For instructions see: https://journals.plos.org/plosone/s/submission-guidelines#loc-laboratory-protocols. Additionally, PLOS ONE offers an option for publishing peer-reviewed Lab Protocol articles, which describe protocols hosted on protocols.io. Read more information on sharing protocols at https://plos.org/protocols?utm_medium=editorial-email&utm_source=authorletters&utm_campaign=protocols.

We look forward to receiving your revised manuscript.

Kind regards,

Richard Rowe

Academic Editor

PLOS ONE

Journal Requirements:

Reviewers' comments:

Reviewer's Responses to Questions

**Comments to the Author**

1. Is the manuscript technically sound, and do the data support the conclusions?

Reviewer #1: Yes

Reviewer #2: Yes

2. Has the statistical analysis been performed appropriately and rigorously? 

Reviewer #1: Yes

Reviewer #2: Yes

3. Have the authors made all data underlying the findings in their manuscript fully available?

Reviewer #1: Yes

Reviewer #2: Yes

4. Is the manuscript presented in an intelligible fashion and written in standard English?

Reviewer #1: Yes

Reviewer #2: Yes

5. Review Comments to the Author

Reviewer #1: The study presents data on 3948 participants including 478 male and 668 female MZ paired twins, 302 male and 410 DZ paired same-sex twins, and 343 male and 343 female opposite sex paired twins, plus single twins and singletons. In 2012-2015 follow-up, depressive symptoms were measured by BDI-II and 8 domains of health-related quality of life (HRQL) was measured by Ware's SF-36. Main findings are that depressive symptoms are negatively correlated with all the HRQF measures as expected in studies in West. Assuming an ACE model, estimates of heritability were significant only for depressive symptoms (33% A and 67% E) and emotional well-being (23% A and 87% E). C was prominent in HRQL for women, but not in men. E was strong for all traits (67% to 93%). For the heritable trait of depressive symptoms, there were weak but significant phenotypic correlations with emotional role functioning, which is not significantly heritable in either sex, (-.11 men and -.18 women) and emotional well-being (-.23), consistent with overlap in the measures and/or weak genetic overlap.

The writing and analytic work is excellent. The modeling of possible sex differences considered alternative models of no sex difference or a difference by a scalar multiplier. This was well done.

Another strength is that the evidence for shared environmental effects in women is reasonably interpretable in terms of the psychosocial and socioeconomic conditions of women in Sri Lanka. This paper provides a good example of the utility of research in different cultures.

A limitation that should be discussed is the assumption of the ACE model itself when studying complex phenotypes like health-related quality of life. Recent studies of the genetics of personality and its strong relationship to health and well-being indicate that there are extensive gene-gene and gene-environmental effects in health-related well-being (Zwir I, et al, Molecular Psychiatry 2020-2021). This calls linear ACE models into question because the relevant genes and/or environments are not really functioning independently. Despite this general concern, the pattern of twin correlations observed are in fact more consistent with an ACE model rather than an epistatic or g-e model (e.g., none have MZ >> DZ correlations), so use of an ACE model is apparently adequate for the data examined. Nevertheless, the description of E in the discussion as involving lifestyle, life events, or chronic illness needs to be more nuanced because personality traits with complex inheritance patterns are strong, but partial, determinants of all these variables (Cloninger et al, Translational Psychiatry 9:290; https://rdcu.be/bZbaJ). This has important implications also for the future directions because it would be inadequate to consider the direction of effects between depressive symptoms and HRQL without recognizing that they both develop in the more-or-less self-regulated lives of people in a complex psychosocial, sociocultural, and ecological context. For example, to understand how to improve the lives of women in Sri Lanka effectively will require a comprehensive understanding of how their social and economic position might be enhanced so that they can function with more psychological well-being. Considering BDI and SF-36 scores in a vacuum is unlikely to be very revealing, and the Colombo study may well have the potential to shed light on this.

Reviewer #2: I really enjoyed the paper. The methodology used is of an excellent level. I have very few suggestions.

In general, I recommend a simpler exposition that allows readers, without high skills in Behavioral Genetics, to understand the rationale of the analyses. I have nothing to underline with regard to the methodology and the analyzes which, as I said, are absolutely adequate for the objectives and purposes of the work.

Introduction

I suggest to better explain the importance and the rationale to investigate the common etiological factors under the covariation between depressive symptoms and poor HRQL.

Methods

A better explanation of the models should be provided. For example, authors reported ‘the female variance is multiplied by a constant (scalar)’

What does it mean?

Results

The reader who is not expert in BG could have problems in interpreting the model-fitting results.

What does it mean that the best model is the scalar one? what repercussions does it have on the knowledge of the covariance between the two phenotypes?

Discussion

Discussion could be benefit from a clinical interpretation of the data.

6. PLOS authors have the option to publish the peer review history of their article (what does this mean?). If published, this will include your full peer review and any attached files.

Reviewer #1: No

Reviewer #2: No

---

## [Author Response · Author response to Decision Letter 0]

18 Feb 2022

EDITOR COMMENTS

1. Abstract

i) Results: The quoted figures for depression add up to 100%, for HRQL, a range is given for E and a combined range for A and C. It is not clear in abstract what the ranges are: confidence intervals? Sub-scales? I recommend separating out the A and C estimates. I am not sure how they can add up to 100% with the ranges given either. For example, if the E estimate is at the upper range, should there be a lower range value for A and C to keep the total to 100%?

Thank you for your observations. The A and C estimates represent all subscales of HRQL (8 domains), therefore a range is given. The estimates of A, C, and E add up to 100% for the individual HRQL variables (as you can see on Table 4, page 17, line 373). In the abstract however, we only gave the significant results for A and C, which ranged from 22% to 28%, therefore the estimates would not add up to 100%. 

We have now split the A and C estimates and gave a range that includes non-significant results (A= 0-23%; C=0-28%). We also included the word “subscales” to indicate what those ranges refer to. A sentence was added that refers to the results that were significant (A=23%; C=22-28%).

Therefore, the relevant section in the abstract (Page 3, lines 52-57) has been modified as follows:

“Nonshared environment accounted for the majority of variance in all the subscales of HRQL (ranging from 68 to 93%), alongside small genetic influences (ranging from 0 to 23%) and shared environmental influences (ranging from 0 to 28%). Genetic influences were significant for emotional wellbeing (23%). Shared environmental influences were significant for four out of the eight HRQL variables (ranging from 22-28%), and they were more prominent in females than males. “

ii)Rph-A etc abbreviations needs explanation for this journal as you give in the method. That won’t help readers at the abstract stage, however. I recommend leaving these figures out of the abstract and just explaining conceptually what these results mean here.

We agree. Abbreviations Rph-A etc and the values were removed from the manuscript. The concepts were already described in lines 58-59, e.g. “These correlations were mostly explained by overlapping nonshared environmental effects.”

2. Introduction

Final paragraph. Check numbering.

Page 7, line 157: Numbering changed from iii to ii.

3. Method

Genetic Model fitting: Please add a reference supporting the validity of the twin modelling assumptions mentioned.

Page 11, line 240: References no 39 and 40 were added to the text and to the reference list.

“39. Plomin R, DeFries J, Knoipik V, Neiderhiser J. Behavioral Genetics. 6th ed. New York: Worth Publishers; 2013.”

40. Kendler KS, Neale MC, Kessler RC, Heath AC, Eaves LJ. A test of the equal-environment assumption in twin studies of psychiatric illness. Behav Genet. 1993 Jan; 23(1):21-7.”

4. Discussion 

Para 1: Please comment on why the results of this study may differ from other analyses of the same dataset.

Page 22, lines 485-486: A sentence was added to explain this: 

“Fitting a model with the same A, C and E parameters and allowing the female variance to be a scalar bigger resulted in a better fitting (more parsimonious) model.” 

5. Journal requirements

i)Please ensure that your manuscript meets PLOS ONE's style requirements, including those for file naming. 

Line numbering was added

On title page, lines 23-24, we have included an additional affiliation for one of the authors. 

Page 11, line 231: Paragraph title (“Genetic model fitting”) was reformatted -> 14pt to 16pt.

Page 21, lines 458-460: Font on Table 5 legend was reformatted -> 11pt to 12pt.

Pages 34-35, lines 780-829: Bold font was removed from the Supporting tables’ captions.

ii)Please ensure your reference list is complete and correct. If you have cited papers that have been retracted, please include the rationale for doing so in the manuscript text, or remove these references and replace them with relevant current references. Any changes to the reference list should be mentioned in the rebuttal letter that accompanies your revised manuscript. If you need to cite a retracted article, indicate the article’s retracted status in the References list and also include a citation and full reference for the retraction notice.

Thank you for your observations. Reference list has been updated, as well as the in-text citation numbering throughout the manuscript.

Specifically, we have added three new references [39, 40, and 45] and changed the numerical order in the list and in all in-text areas where the numbering did not reflect the correct order [39-42].

REVIEWER #1

The study presents data on 3948 participants including 478 male and 668 female MZ paired twins, 302 male and 410 DZ paired same-sex twins, and 343 male and 343 female opposite sex paired twins, plus single twins and singletons. In 2012-2015 follow-up, depressive symptoms were measured by BDI-II and 8 domains of health-related quality of life (HRQL) was measured by Ware's SF-36. Main findings are that depressive symptoms are negatively correlated with all the HRQF measures as expected in studies in West. Assuming an ACE model, estimates of heritability were significant only for depressive symptoms (33% A and 67% E) and emotional well-being (23% A and 87% E). C was prominent in HRQL for women, but not in men. E was strong for all traits (67% to 93%). For the heritable trait of depressive symptoms, there were weak but significant phenotypic correlations with emotional role functioning, which is not significantly heritable in either sex, (-.11 men and -.18 women) and emotional well-being (-.23), consistent with overlap in the measures and/or weak genetic overlap. 

The writing and analytic work is excellent. The modeling of possible sex differences considered alternative models of no sex difference or a difference by a scalar multiplier. This was well done. Another strength is that the evidence for shared environmental effects in women is reasonably interpretable in terms of the psychosocial and socioeconomic conditions of women in Sri Lanka. This paper provides a good example of the utility of research in different cultures.

Thank you.

1.Discussion 

A limitation that should be discussed is the assumption of the ACE model itself when studying complex phenotypes like health-related quality of life. Recent studies of the genetics of personality and its strong relationship to health and well-being indicate that there are extensive gene-gene and gene-environmental effects in health-related well-being (Zwir I, et al, Molecular Psychiatry 2020-2021). This calls linear ACE models into question because the relevant genes and/or environments are not really functioning independently. Despite this general concern, the pattern of twin correlations observed are in fact more consistent with an ACE model rather than an epistatic or g-e model (e.g., none have MZ >> DZ correlations), so use of an ACE model is apparently adequate for the data examined.

Nevertheless, the description of E in the discussion as involving lifestyle, life events, or chronic illness needs to be more nuanced because personality traits with complex inheritance patterns are strong, but partial, determinants of all these variables (Cloninger et al, Translational Psychiatry 9:290; https://rdcu.be/bZbaJ). This has important implications also for the future directions because it would be inadequate to consider the direction of effects between depressive symptoms and HRQL without recognizing that they both develop in the more-or-less self-regulated lives of people in a complex psychosocial, sociocultural, and ecological context. For example, to understand how to improve the lives of women in Sri Lanka effectively will require a comprehensive understanding of how their social and economic position might be enhanced so that they can function with more psychological well-being. Considering BDI and SF-36 scores in a vacuum is unlikely to be very revealing, and the Colombo study may well have the potential to shed light on this.

These are all very fair points, thank you for the suggestions. On Page 26, lines 592-596, we have added a comment to the “future directions” paragraph to include your suggested limitations: 

“Still, temperamental characteristics, that are strongly influenced by genetics, might exert effects on both physical and emotional wellbeing, as well as on the nonshared environment. For example, higher scores in pessimism could increase the impact of poverty on an individual. Such additional factors should be controlled when trying to infer directionality.”

Also, reference to the Cloninger et al study was included in this section and added to the reference list on Page 34, lines 765-767.

“45. Cloninger CR, Cloninger KM, Zwir I, Keltikangas-Järvinen L. The complex genetics and biology of human temperament: a review of traditional concepts in relation to new molecular findings. Transl Psychiatry. 2019 Dec;9(1):290.”

REVIEWER #2

I really enjoyed the paper. The methodology used is of an excellent level. I have very few suggestions. In general, I recommend a simpler exposition that allows readers, without high skills in Behavioral Genetics, to understand the rationale of the analyses. I have nothing to underline with regard to the methodology and the analyzes which, as I said, are absolutely adequate for the objectives and purposes of the work.

Thank you.

1.Introduction

I suggest to better explain the importance and the rationale to investigate the common etiological factors under the covariation between depressive symptoms and poor HRQL.

We have added the following rationale at the end of the Introduction, page 7, lines 159-162.

“The importance of studying the common aetiology between depression and HRQL is to gain insight into the mechanisms of depression that may contribute to our understanding and aid development of therapeutic interventions and prevention strategies to reduce mental health as well as related quality of life”.

2.Method

 A better explanation of the models should be provided. For example, authors reported ‘the female variance is multiplied by a constant (scalar)’. What does it mean?

We have added a better explanation of the model on Page 12, lines 267-272, as follows:

“In addition to the models above, we have to rule out that sex differences show up due to absolute variance differences rather than aetiology. For example: A=60, C=20 and E=20 in females and A=6, C=2 and E=2 in males in both cases give the same standardized results. This is modelled by estimating the same A, C and E for males and females but allowing the variance for females to be multiplied by a constant to scale up the variance”

3.Results

The reader who is not expert in BG could have problems in interpreting the model-fitting results. What does it mean that the best model is the scalar one? what repercussions does it have on the knowledge of the covariance between the two phenotypes?

Page 16, lines 332-339: Variance decomposition: We totally agree that for the non-BG audience it is not relevant which model is best fitting (these are reported, and selection justified in detail in S4 Table for the experts). Essential is what the selected model means. This is mainly to conclude whether there is evidence for sex differences or not in aetiology. We have therefore slightly re-written the text so that the meaning of the model is focused on, rather than the selected/fitted models which may be distracting. We have carried this forward to description of the Covariance decomposition section.

4.Discussion

Discussion could benefit from a clinical interpretation of the data.

Page 27, lines 605-607: A sentence was added to the “Future directions” paragraph: 

“Identifying the nature of individual-level environmental exposures could also improve person-centred therapeutic interventions to reduce mental and physical health decline.”

---

## [Editor Report · Decision Letter 1]

2 Mar 2022

The aetiological relationship between depressive symptoms and health-related quality of life: A population-based twin study in Sri Lanka

PONE-D-21-28109R1

Dear Dr. Triantafyllou,

We’re pleased to inform you that your manuscript has been judged scientifically suitable for publication and will be formally accepted for publication once it meets all outstanding technical requirements. Very many thanks for your careful response to the reviewers' comments.

Kind regards,

Richard Rowe

Academic Editor

PLOS ONE
---

## [Editor Report · Acceptance letter]

7 Mar 2022

PONE-D-21-28109R1 

The aetiological relationship between depressive symptoms and health-related quality of life: A population-based twin study in Sri Lanka 

Dear Dr. Triantafyllou:

I'm pleased to inform you that your manuscript has been deemed suitable for publication in PLOS ONE. Congratulations! Your manuscript is now with our production department. 

Kind regards, 

on behalf of

Professor Richard Rowe 

Academic Editor

PLOS ONE